# Ultrasensitive Simultaneous Detection of Multiple Rare Modified Nucleosides as Promising Biomarkers in Low-Put Breast Cancer DNA Samples for Clinical Multi-Dimensional Diagnosis

**DOI:** 10.3390/molecules27207041

**Published:** 2022-10-19

**Authors:** Yue Yu, Hui-Yu Pan, Xin Zheng, Fang Yuan, Ying-Lin Zhou, Xin-Xiang Zhang

**Affiliations:** 1Beijing National Laboratory for Molecular Sciences (BNLMS), MOE Key Laboratory of Bioorganic Chemistry and Molecular Engineering, College of Chemistry and Molecular Engineering, Peking University, Beijing 100871, China; 2Institute of Biotechnology Development, Qilu Pharmaceutical, Jinan 250100, China; 3Beijing Key Laboratory of Clinical PK & PD Investigation for Innovative Drugs, Clinical Pharmacology Research Center, Peking Union Medical College Hospital, Chinese Academy of Medical Sciences & Peking Union Medical College, Beijing 100032, China; 4National Institute of Measurement and Testing Technology, Chengdu 610021, China

**Keywords:** modified nucleosides, chemical labeling, LC-MS/MS

## Abstract

Early cancer diagnosis is essential for successful treatment and prognosis, and modified nucleosides have attracted widespread attention as a promising group of cancer biomarkers. However, analyzing these modified nucleosides with an extremely low abundance is a great challenge, especially analyzing multiple modified nucleosides with a different abundance simultaneously. In this work, an ultrasensitive quantification method based on chemical labeling, coupled with LC-MS/MS analysis, was established for the simultaneous quantification of 5hmdC, 5fdC, 5hmdU and 5fdU. Additionally, the contents of 5mdC and canonical nucleosides could be obtained at the same time. Upon derivatization, the detection sensitivities of 5hmdC, 5fdC, 5hmdU and 5fdU were dramatically enhanced by several hundred times. The established method was further applied to the simultaneous detection of nine nucleosides with different abundances in about 2 μg genomic DNA of breast tissues from 20 breast cancer patients. The DNA consumption was less than other overall reported quantification methods, thereby providing an opportunity to monitor rare, modified nucleosides in precious samples and biology processes that could not be investigated before. The contents of 5hmdC, 5hmdU and 5fdU in tumor tissues and normal tissues adjacent to the tumor were significantly changed, indicating that these three modified nucleosides may play certain roles in the formation and development of tumors and be potential cancer biomarkers. While the detection rates of 5hmdC, 5hmdU and 5fdU alone as a biomarker for breast cancer samples were 95%, 75% and 85%, respectively, by detecting these three cancer biomarkers simultaneously, two of the three were 100% consistent with the overall trend. Therefore, simultaneous detection of multiple cancer biomarkers in clinical samples greatly improved the accuracy of cancer diagnosis, indicating that our method has great application potential in clinical multidimensional diagnosis.

## 1. Introduction

Diagnosis of cancer at an early stage is critical for successful treatment and patient prognosis, and the study of new biomarkers for early cancer diagnosis is currently undergoing rapid development [1]. Recently, the nucleosides and their modified forms have been considered to be a promising group of early cancer biomarkers, drawing broad attention from biologists and chemists [2,3,4]. Studies have shown that DNA methylation patterns are globally disrupted in cancer, with a genome-wide decrease of 5-methyl-2′-deoxycytidine (5mdC) and a gene-specific increase of 5mdC occurring simultaneously in the same cell [5]. Moreover, the levels of 5-hydroxymethyl-2′-deoxycytidine (5hmdC) are dramatically reduced in a broad spectrum of cancers associated with the substantial reduction in the expression of three ten–eleven translocation (TET) genes when compared to the matched surrounding normal tissues, thus suggesting the critical roles of aberrant DNA demethylation for the oncogenic process [6,7,8]. 5-Formyl-2′-deoxycytidine (5fdC) was also proved to have more functional roles in tumor formation and development than the intermediate in the DNA demethylation pathway [9,10]. In addition to these cytidine modifications, thymidine modifications, such as 5-hydroxymethyl-2′-deoxyuridine (5hmdU) and 5-formyl-2′-deoxyuridine (5fdU), have been discovered to be recognized by transcription factors recently, suggesting their potential roles in gene regulation related to cancer [9,11,12]. Therefore, the detection of these modified nucleosides considered as cancer biomarkers might have important guiding significance in the early diagnosis, treatment and prognosis of tumors.

The simultaneous detection of multiple cancer biomarkers in clinical samples can provide more information about the formation and development of cancer, which is conducive to improving the accuracy of cancer diagnosis [13,14]. Similarly, the multiple detection of modified nucleosides, which may be promising diagnostic biomarkers, is also of great significance. However, modified nucleosides differ in their abundance within the genome. The content of 5mdC accounts for approximately 2–7% of all cytidines in mammals, and 5hmdC is present at a frequency of approximately 10- to 100-fold lower than that of 5mdC [15,16,17]. In particular, the contents of 5fdC, 5fdU and 5hmdU are much lower than those of 5mdC. The 5fdC level is about 1–20 per 10^6^ cytidines in mammals, and the contents of 5hmdU and 5fdU in mammalian tissue samples are 0.1–30 per 10^6^ nucleosides [9,18,19]. It is a great challenge to analyze these modified nucleosides with extremely low abundance, especially to analyze multiple modified nucleosides with a different abundance simultaneously.

Thus far, multiple methods have been developed for the quantification of these potential biomarkers [20]. Some methods based on immunology [21,22], electrochemistry [23,24] and surface-enhanced raman scattering [25] have advantages in detection sensitivity. However, they usually cannot realize the simultaneous detection of multiple modified nucleosides, and thus are still lacking in universality. Detection methods based on separation technologies are more common in the overall analysis of modified nucleosides, and liquid chromatography, coupled with tandem mass spectrometry (LC-MS/MS), has been the most common method for the overall analysis of modified nucleosides due to its inherent selectivity, sensitivity and stability [9,17,18,19,26]. To further increase the detection sensitivities for modified nucleosides, chemical derivatization was adopted to introduce an easily ionizable moiety. Ultrasensitive methods for the quantification of 5hmdC [27], 5fdC and 5-carboxyl-2′-deoxycytidine 5cadC [28] have been established, respectively, based on the labeling of reagents designed and synthesized by our lab [29]. The detection sensitivities of these rare modified cytosines were improved by 100–178 times upon derivatization. Therefore, we hope to extend the hydrazine–triazine based labeling reactions to the analysis of more low-abundance modified nucleosides in order to establish a method for the simultaneous detection of multiple rare modified nucleosides.

In this study, an ultrasensitive method for the simultaneous determination of nine nucleosides, including four rare modified nucleosides, namely 5hmdC, 5fdC, 5hmdU and 5fdU, in limited biological samples, was established based on chemical labeling coupled with the LC-MS/MS system. The detection sensitivities of 5hmdU and 5fdU were greatly improved, by 275-fold and 850-fold, as compared to the unlabeled nucleosides under optimal conditions, making it the most sensitive genome-wide overall quantification method. With this method, we achieved the simultaneous detection of nine nucleosides with different abundances in about 2 μg genomic DNA of breast tissues from 20 breast cancer patients. The results showed that the contents of 5hmdC, 5hmdU and 5fdU in tumor tissues and normal tissues adjacent to the tumor were significantly changed, and the simultaneous detection of multiple modified nucleosides could greatly improve the accuracy of cancer diagnosis, indicating that our method had great advantages in clinical, multi-dimensional diagnosis.

## 2. Results and Discussion

### 2.1. The Establishment of Ultrasensitive Simultaneous Detection of Multiple Rare Modified Nucleosides Based on Chemical Labeling

Due to the extremely low abundance and the low ionization efficiency in regular ESI of 5hmdC, 5fdC, 5hmdU and 5fdU, the quantification of these rare modified mucleosides in limited biological and clinical samples is still challenging, which greatly affects the study of their biological function. In our previous work, a hydrazine–triazine based labeling reagent i-Pr_2_N, designed and synthesized by our lab, had shown excellent performance in the chemical labeling and sensitive detection of 5hmdC and 5fdC based on HPLC-MS/MS [27,28]. Considering the similar structure of 5hmdU to 5hmdC and 5fdU to 5fdC, we expected that, after labeling i-Pr_2_N, 5hmdU and 5fdU would obtain satisfactory sensitivity. In the design of the chemical labeling strategy based on i-Pr_2_N, 5fdC and 5fdU were directly derivatized by i-Pr_2_N for the quantification, due to the easy reaction between the formyl group and hydrazide group under mild conditions (Figure 1A), and 5hmdC and 5hmdU were first oxidized to 5fdC and 5fdU using MnO_2_ and then derivatized by i-Pr_2_N for the quantification (Figure 1B). Therefore, ultrasensitive simultaneous detection of four rare, modified nucleosides, 5mdC and canonical nucleosides were obtained by one-step and two-step reactions, respectively.

Since the performance of 5hmdC and 5fdC, based on i-Pr_2_N labeling, has previously been investigated [27,28], we focused on the performance of 5hmdU and 5fdU upon derivatization. First, to reach the maximal chemical labeling efficiency, we optimized the oxidation and labeling reaction conditions. We started by optimizing the reaction conditions using nucleosides standards. LC-MS/MS was used to monitor the reaction. The oxidation conditions, including reaction temperature, time, MnO_2_ amount and FA content were first optimized. The reaction temperature was optimized, ranging from 30 °C to 70 °C, and the results showed that 50 °C was the best temperature for the reaction (Appendix A). To optimize oxidation time, the same reagents were mixed together and incubated at 50 °C for 20 min to 2 h. Appendix A showed that 1 h was enough for the oxidation. MnO_2_ amount was also optimized from 1 mg to 10 mg. Appendix A showed that 5 mg MnO_2_ was enough for the oxidation. As for the optimization of the FA content, the results showed that the addition of FA could increase the reaction efficiency, and the peak area of the derivatives reached a plateau when the content of FA was 2 μL (Appendix A). Taken together, the optimized reaction conditions for 5hmdU oxidation by MnO_2_ were under 50 °C for 1 h using 5 mg MnO_2_ and 2 μL FA.

Then, we examined the labeled products of 5fdU using i-Pr_2_N. As expected, 5fdU successfully reacted with i-Pr_2_N and formed the desired derivatives. The product ion spectrum showed that m/z 548.4 and 432.2, which represented the parent ion of the 5fdU-i-Pr_2_N and its product ion, were observed after labeling (Appendix A). The derivatization conditions, including reaction time, concentration of i-Pr_2_N and HAc content, were optimized. The reaction time, ranging from 0 to 30 min for the derivatization reaction, was first optimized. The results showed that this reaction could only achieve extremely high derivatization efficiency by votexing without additional reaction time at room temperature (Appendix A). The optimal concentrations of i-Pr_2_N and HAc content were investigated. The results showed that 1 mg mL^−1^ i-Pr_2_N and 2 μL HAc were enough for the reaction (Appendix A). Taken together, the optimized derivatization conditions for 5fdU were vortex stirring at room temperature using 1 mg mL^−1^ i-Pr_2_N and 2 μL HAc.

The main purpose for chemical labeling is to increase the detection sensitivities of the analytes during LC-MS/MS analysis. We then compared the limits of detection (LODs) of 5hmdU and 5fdU upon labeling by i-Pr_2_N. The extracted ion chromatograms of unlabeled and i-Pr_2_N-labeled products were shown in Figure 2. The results showed that the signal intensities of 1 nM 5hmdU-i-Pr_2_N and 5fdU-i-Pr_2_N were much higher than those of 50 nM 5hmdU and 5fdU, demonstrating that chemical labeling significantly improved the MS detection sensitivities of 5hmdU and 5fdU. The LODs, defined as the amounts of the analytes at a signal-to-noise ratio (S/N) of three, were employed to evaluate the improved LODs of 5hmdU and 5fdU by labeling with i-Pr_2_N. As shown in Appendix A, the chemical labeling dramatically increased the detection sensitivities of 5hmdU by 275-fold and 5fdU by 850-fold compared with the unlabeled nucleosides. Without enrichment, the LODs of the 5hmdU and 5fdU derivatives upon i-Pr_2_N were as low as 44.4 amol and 26.0 amol respectively, which was the most sensitive LC-MS/MS based genome-wide overall quantification method for these nucleosides [9,18,26,30,31,32] (Appendix A). There are two main reasons for the significant increase in the detection sensitivity of chemical derivatization methods. First, an easily ionizable group was introduced to increase the ionization efficiency. Second, the hydrophobicity of the analytes was dramatically increased, which can finally increase the detection sensitivities according to the mechanism of ESI of small molecules and higher desolvation efficiency under higher organic content in LC-MS/MS analysis [33,34,35,36].

The performance of the method for the quantification of 5hmdU and 5fdU were evaluated. We used the mean peak areas of 5hmdU-i-Pr_2_N and 5fdU-i-Pr_2_N as the ordinate and the actual molars of 5hmdU and 5fdU as the abscissa to construct the calibration curves. The results showed that the linear correlation coefficients of 5hmdU-i-Pr_2_N and 5fdU-i-Pr_2_N were greater than 0.997 within the range of 10 pM to 1000 pM, which proved that the method had good linearities and a wide linear range (Appendix A). Additionally, the calibration curves of canonical nucleosides and 5mdC were constructed by plotting the mean peak area versus their actual molar, and the calibration curves of 5hmdC and 5fdC were constructed by plotting the mean peak area of 5fdC-i-Pr_2_N versus their actual molar (Appendix A). As 5hmdU and 5fdU were kinds of oxidized thymine nucleosides, we then studied the oxidation of thymidine (dT) under our reaction conditions. The results showed that a little bit of dT was oxidized to 5fdU, which affected the quantitative results. We found that there was a linear relationship between the yield of 5fdU and the content of dT. Therefore, the calibration curve of dT oxidation was constructed by plotting the mean peak area of 5fdU-i-Pr_2_N after oxidized dT by MnO_2_ and labeled by i-Pr_2_N versus the actual molar of dT (Appendix A).

In addition, we added different amounts of 5hmdU and 5fdU standards to real samples, and compared the measured values detected by the developed method with the theoretical values to investigate the accuracy and precision of the method. As shown in Appendix A, the recovery was from 90.8% to 112.7%, and the intra-day and inter-day relative standard deviations (RSDs) were less than 7.8%, indicating that the method has good reproducibility and accuracy.

### 2.2. Contents of Modified Cytosines in Human Breast Cancer Tissues

Previous studies showed that aberrant epigenetic modifications were closely related to cancer [37,38]. Whether there were differences in 5mdC, 5hmdC, 5fdC, 5hmdU and 5fdU in tumor tissues was explored. Human breast cancer was chosen as the model, and genomic DNA (gDNA) samples were extracted from 20 pairs of breast cancer tissues and the corresponding normal tissues adjacent to the tumor for the quantification of these five modified nucleosides (Figure 3). Each gDNA sample after enzymatic hydrolysis was divided into two equal parts. One was directly proceeded to labeling step for the quantification of 5fdC and 5fdU, and further diluted 10 times for the analysis of canonical nucleosides and 5mdC. The other proceeded to the oxidation step and labeling step for the quantification of 5hmdC and 5hmdU. The minimum amount of extracted gDNA was only approximately 2 μg, which could not be detected by other reported genome-wide overall quantification methods, thus showing the advantages of our method in clinical sample detection. The contents of these modified nucleosides are shown in Figure 4 and Appendix A.

The quantification results showed that the contents of 5mdC (*p* < 0.1), 5hmdC (*p* < 0.01) and 5fdC (*p* < 0.1) decreased, while the contents of 5hmdU (*p* < 0.05) and 5fdU (*p* < 0.01) increased in tumor tissues compared to normal tissues adjacent to the tumor in general. The changes in 5hmdC, 5hmdU and 5fdU contents were particularly significant compared to 5mdC level and 5fdC level, thus demonstrating the great potential of 5hmdC, 5hmdU and 5fdU as a cancer biomarker. Specifically, for 5hmdC, only the content of 5hmdC in patient No. 14 did not conform to the overall trend, but the contents of other two modified nucleosides 5hmdU and 5fdU did. For 5hmdU, the content of 5hmdU in five patients (No. 1, 4, 8, 9 and 12) was not in line with the overall trend. However, the contents of the other two modified nucleosides 5hmdC and 5fdU in these five patients were in line with the overall trend. And for 5fdU, although the content of 5fdU in three patients (No. 5, 6 and 16) did not conform to the change, the contents of the other two modified nucleosides 5hmdC and 5hmdU in these three patients did. Therefore, as long as the contents of the two cancer biomarkers conformed to the overall trends, we could make a preliminary judgment on the state of the tissues. This meant that we could distinguish 100% between cancer and adjacent tissues by simultaneous detection of these three cancer biomarkers in this work. To our knowledge, the current staging standard of breast tumours usually uses the TNM staging method, which is widely used around the world. Since the stage-related information was not involved, we were unable to compare our results with the parameters of the stage. However, it is worth noting that this work found significant changes in the contents of 5hmdC, 5hmdU and 5fdU in breast cancer tissues and adjacent tissues, thus demonstrating the great potential of 5hmdC, 5hmdU and 5fdU as cancer biomarkers. Additionally, the results indicated that simultaneous detection of multiple cancer biomarkers could provide more information on the tumor development and improve the accuracy of cancer diagnosis. With our method, several rare, modified nucleosides could be detected simultaneously, which had great application potential in clinical multi-dimensional diagnosis. In the next work, more clinical samples from more breast cancer patients and different types of cancer patients will be analyzed to further determine whether these modified nucleosides can become new universal cancer biomarkers for clinical application.

## 3. Materials and Methods

### 3.1. Chemicals and Clinical Samples

2′-Deoxyadenosine (dA), thymidine (dT), 2′-deoxycytidine (dC), 2′-deoxyguanosine (dG), 5-methyl-2′-deoxycytidine (5mdC), manganese dioxide (MnO_2_) and Erythro-9-(2-hydroxy-3-nonyl) adenine (EHNA) were purchased from Sigma-Aldrich (Beijing, China). 5-Hydroxymethyl-2′-deoxyuridine (5hmdU), 5-formyl-2′-deoxycytidine (5fdC) and 5-hydroxymethyl-2′-deoxycytidine (5hmdC) were purchased from Berry & Associates (Dexter, MI, USA). Methanol (MeOH) and acetonitrile (ACN) were LC-MS grade, purchased from J.T.Baker. Formic acid (FA, LC-MS grade) was purchased from Tokyo Chemical Industry (Tokyo, Japan). Acetic acid (HAc, 99.9985%) was purchased from Alfa Aesar (Haverhill, MA, USA). Ultrapure water was purified using a 0.45 μm MF-membrane filter (Merck Millpore, Burlington, NJ, USA) was used in this work. A total of 40 tissue samples from breast cancer patients, including 20 pairs of breast cancer tissues and matched normal tissues adjacent to the tumor were collected from Clinical Biobank, Peking Union Medical College Hospital, Chinese Academy of Medical Sciences.

### 3.2. MnO_2_ Oxidation

5hmdU could be oxidized to 5fdU using MnO_2_. In order to achieve the best oxidation efficiency and the highest MS intensity, the oxidation conditions, including reaction temperature, time, MnO_2_ amount and FA content were optimized. All the reactions were performed with 5 nM of 5hmdU. Briefly, 5 μL, 5 nM and 5hmdU was added into 13 μL ACN and 2 μL FA. Then, 10 mg MnO_2_ was added. The mixture was incubated at a temperature ranging from 30 °C to 70 °C for 1 h and centrifuged at 12,000 rpm for 5 min at room temperature. The supernatant was removed and transferred to a clean centrifuge tube. The black precipitate was washed with 100 μL ACN (the ACN was also collected and mixed with the supernatant of the reaction). After drying the result solution, it was reconstituted in 50 μL 1:1 (*v*:*v*) water/ACN and subjected to LC-MS/MS analysis. Then, the same reagents were mixed together and reacted at 50 °C for 20 min to 2 h to optimize the reaction time. The MnO_2_ amount was optimized from 1 mg to 10 mg, and the FA content was also optimized from 0 μL to 4 μL.

### 3.3. Synthesis of 5-Formyl-2′-Deoxyuridine (5fdU)

5FdU was synthesized by oxidation of 5hmdU using MnO_2_. Briefly, 5 μL, 10 mM, 5hmdU and 50 mg MnO_2_ was added into 13 μL ACN and 2 μL FA. The mixture was reacted at 50 °C for 2 h and centrifuged at 12,000 rpm for 5 min at room temperature. The supernatant was removed and transferred to a clean centrifuge tube. The black precipitate was washed with 100 μL ACN, and the ACN was collected and mixed with the reaction supernatant. The result solution was diluted 10 times for LC-MS/MS analysis, and there was no 5hmdU peak, indicating that 5hmdU had been completely oxidized to 5fdU. We completed 20 reactions in parallel and combined the result solutions. The mixture was then dried at 45 °C using a speed vacuum concentrator for weighing, reconstituted in water to 1 mM as a stock solution of 5fdU. We confirmed the synthesized 5fdU standards by high-resolution MS analysis (Q-Exactive Mass Spectrometer, Thermo, Waltham, MA, USA) and the results are shown in Appendix A.

### 3.4. Chemical Labeling

Our lab has designed and synthesized a series of hydrazine–triazine based labeling reagents, which were successfully applied to improve the MS detection sensitivity of N-glycans [29]. In this study, i-Pr_2_N (available through Beijing Omics Biological Technology Co., Ltd., Beijing, China) was used to derivatize 5fdU and 5fdC. We examined the labeled product of 5fdU using i-Pr_2_N. The product ion spectrum was shown in Appendix A, and *m*/*z* 548.4 and 432.2 represented the parent ion of the 5fdU-i-Pr_2_N and its product ion, indicating that 5fdU successfully reacted with i-Pr_2_N and formed the expected derivative. The derivatization conditions, including reaction time, concentration of i-Pr_2_N and HAc content were optimized to obtain the maximal labeling efficiency. All the reactions were performed with 5 nM of 5fdU. Briefly, 5 μL, 5 nM and 5fdU was added into 14 μL ACN and 1 μL HAc with 5 mg mL^−1^ i-Pr_2_N, and the mixture was incubated at 37 °C. The reaction time ranged from 0 to 30 min for the derivatization reaction. The resulting solution was dried and reconstituted in 50 μL 1:1 (*v*:*v*) water/ACN for analysis by LC-MS/MS. Then, the concentration of i-Pr_2_N was optimized from 0.25–5 mg mL^−1^, and the HAc content was also optimized from 0–4 μL.

### 3.5. LC-MS/MS Analysis

Thermo Scientific Dionex Ultimate 3000 HPLC (Thermo, Waltham, MA, USA) coupled with a Triple Quad^TM^ 5500 mass spectrometer (Sciex, Framingham, CA, USA) with an ESI source (Turbo Ionspray) was used for the analysis of the oxidation product of 5hmdU and labeled product of 5fdU. The LC separation was performed on a Zorbax Stablebond Analytical SB-C18 column (2.1 mm × 100 mm, 3.5 μm, Agilent Technologies, Santa Clara, CA, USA) at 35 °C. Water containing 0.0085% FA (*v*/*v*, solvent A) and MeOH containing 0.0085% FA (*v*/*v*, solvent B) was employed as the mobile phase. A gradient of 0–5% B for 6 min, 5–80% B for 0.5 min, 80% B for 5 min and 100% B for 5 min was used. The flow rate of the mobile phase was set at 0.3 mL min^−1^. The mass spectrometry detection was performed under the positive ESI mode. The nucleosides and labeled products were monitored using the multiple reaction monitoring (MRM) mode. The MRM parameters were optimized to achieve maximal detection sensitivity (Appendix A). The values for curtain gas, collision gas, ionspray voltage, ion source gas 1, ion source gas 2, temperature, declustering potential and entrance potential were 20, 8, 5500, 55, 60, 600, 70 and 10, respectively.

### 3.6. DNA Extraction and Enzymatic Digestion

The gDNA samples of tissues were extracted using the TaKaRa MiniBEST Universal Genomic DNA Extraction Kit Ver 5.0 (Takara, iotechnology Co., Ltd., Dalian, China) according to the manufacture’s recommended procedure. The concentration of the purified gDNA, which was extracted from one tissue sample, was determined by Nanodrop 2000 Spectrophotometer (Thermo, Waltham, MA, USA). DNA Degradase PlusTM (Zymo Rearch, Irvine, CA, USA) was used to digest the gDNA into single nucleosides according to the protocol. Generally, gDNA in 86 μL of H_2_O was added into 2 μL of EHNA (7.2 mg mL^−1^), 2 μL DNA Degradase Plus and 10 μL 10X DNA Degradase Reaction Buffer. The mixture (100 μL) was then incubated at 37 °C for 2 h. The mixture was divided into two equal parts (50 μL), and both were concentrated by a speed vacuum concentrator to 5 μL, respectively. One (named Mix DOL) proceeded to the oxidation step and labeling step for the quantification of 5hmdC, 5hmdU and 5fdU, and the other (named Mix DL) was directly proceeded to the labeling step for the quantification of 5fdC and 5fdU.

### 3.7. Determination of Multiple Modified Nucleosides of Biological Samples

Briefly, Mix DL was labeled by i-Pr_2_N, which was performed in 15 μL of MeOH with 1 mg mL^−1^, i-Pr_2_N and 2 μL HAc. The mixture was dried using a speed vacuum concentrator and then reconstituted in 50 μL 1:1 (*v*:*v*) water/ACN. The result solution was analyzed by LC-MS/MS for the quantification of 5fdU and 5fdC. The concentrations of 5fdU and 5fdC were calculated using the calibration curve of 5fdU and 5fdC. Mix DL was further diluted 10 times for the analysis of dA, dC, dT, dG and 5mdC. Mix DOL was first oxidized by 5 mg MnO_2_ under 50 °C for 1 h in a 20 μL reaction containing 2 μL FA and centrifuged at 12,000 rpm for 5 min at room temperature. The supernatant was removed and transferred to a clean centrifuge tube. The black precipitate was washed with 100 μL ACN, and the ACN was collected and mixed with the reaction supernatant. The mixture was then concentrated by speed vacuum concentrator to 5 μL and labeled by i-Pr_2_N using the same conditions as mentioned above. The result solution was analyzed by LC-MS/MS for the quantification of 5hmdU and 5hmdC. The concentration of 5hmdC was calculated by the calibration curve of 5hmdC. Compared with the peak area of 5fdU-i-Pr_2_N in Mix DL, the increased part was due to the contribution from 5hmdU and the oxidation of dT. Therefore, the peak area of 5fdU-i-Pr_2_N in Mix DOL minus the peak area of 5fdU-i-Pr_2_N obtained by dT oxidation and the peak area of 5fdU-i-Pr_2_N in Mix DL was the peak area caused by 5hmdU. The concentration of 5hmdU was calculated using the increased peak area by the calibration curve of 5hmdU. Detailed parameters of LC-MS/MS system were the same as mentioned above.

## 4. Conclusions

In conclusion, an ultrasensitive method was established for the quantification of four rare, modified nucleosides, namely 5hmdC, 5fdC, 5hmdU and 5fdU in limited, biological samples. The detection sensitivities of 5hmdC, 5fdC, 5hmdU and 5fdU were dramatically enhanced based on chemical labeling, providing a powerful tool to explore the functions of these modified nucleosides in precious samples and biological processes. With this method, we achieved the simultaneous detection of nine nucleosides in gDNA of breast tissues from 20 breast cancer patients. The results showed that the contents of 5hmdC, 5hmdU and 5fdU in cancer tissues and normal tissues adjacent to the tumor significantly changed, indicating that 5hmdC, 5hmdU and 5fdU might play certain roles in the tumor and might serve as indicators for cancer prognostics. Compared to the detection of single modified nucleoside, simultaneous detection of multiple modified nucleosides could greatly improve the accuracy of cancer diagnosis, indicating great potential for the multi-dimensional diagnosis of cancer. In the next work, more clinical samples from more breast cancer patients and different types of cancer patients will be analysed to further determine whether these modified nucleosides can become new universal cancer biomarkers for clinical application, and further verify the advantages of multiple detection methods for modified nucleosides in clinical applications.

## Figures and Tables

**Figure 1 molecules-27-07041-f001:**
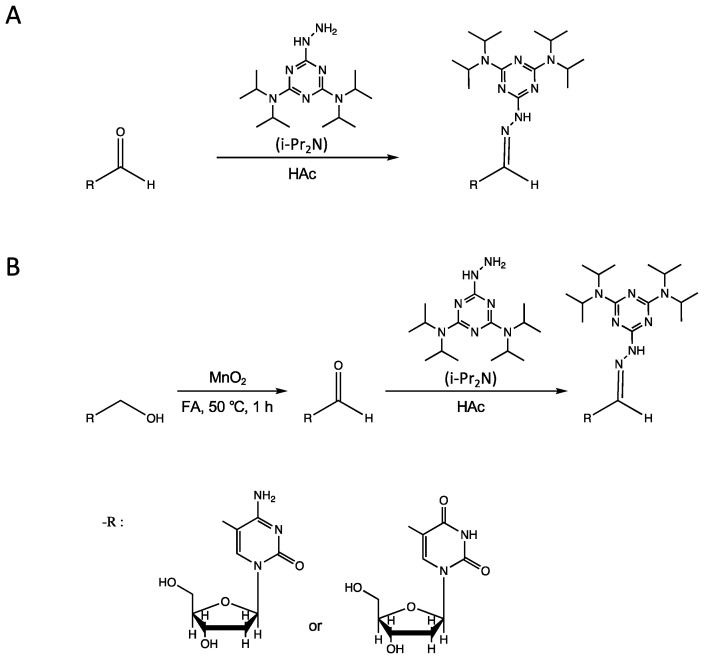
The strategy of chemical labeling of (**A**) 5fdC and 5fdU and (**B**) 5hmdC and 5hmdU with i-Pr_2_N.

**Figure 2 molecules-27-07041-f002:**
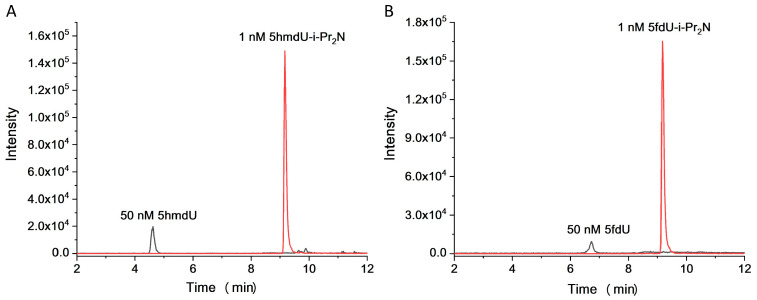
Extracted ion chromatograms of unlabeled and i-Pr_2_N-labeled products of (**A**) 5hmdU and (**B**) 5fdU. The amounts of unlabeled 5hmdU and 5fdU were 50 nM (250 fmol), and the amounts of i-Pr_2_N-labeled products were 1 nM (5 fmol), respectively.

**Figure 3 molecules-27-07041-f003:**
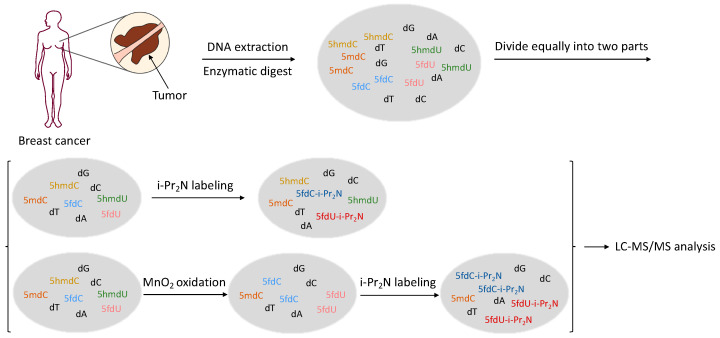
Scheme of practical sample analysis.

**Figure 4 molecules-27-07041-f004:**
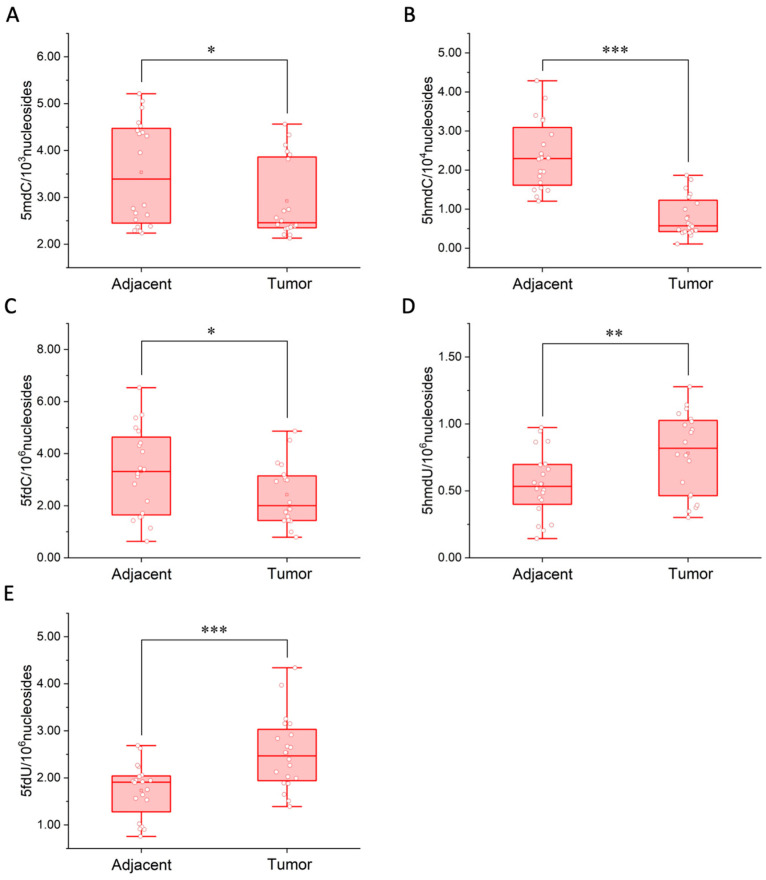
Quantification of (**A**) 5mdC, (B) 5hmdC, (**C**) 5fdC, (**D**) 5hmdU and (**E**) 5fdU in breast cancer tissues. * *p* < 0.1; ** *p* < 0.05; *** *p* < 0.01.

## Data Availability

Data are contained within the article or Appendix A.

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
