# Peer review of "Ultrasensitive Simultaneous Detection of Multiple Rare Modified Nucleosides as Promising Biomarkers in Low-Put Breast Cancer DNA Samples for Clinical Multi-Dimensional Diagnosis"

_molecules, 2022, doi:10.3390/molecules27207041_

Round 1

Reviewer 1 Report

The present paper continues the authors’ series of contributions regarding development of quantification methods for modified nucleosides. The authors has designed a method for determination of  5hmdU and 5fdU and used the new and previously developed methods for the simultaneous determination of modified cytosines and modified uraciles in human breast cancer tissues. As a result, it was concluded that the analysis of several modified nucleosides signifcantly improves the accuracy of cancer diagnosis.

No specific areas of weakness regarding scientific content have been found.

The manuscript is relevant for the field, clearly written and well-structured.

All the references cited in the manuscript are relevant with more than half of them being published recently. The number of self-citing publications is appropriate, no excess is observed.

The manuscript is scientifically sound and the used methods are appropriate for the set goals.

The methods section is written in detail so that the results can be reproduced.

The conclusions are consistent with the presented data and arguments.

No specific comments.

Author Response

Thanks for your comments.

Reviewer 2 Report

Yu and coll have set up an ultrasensitive  method to investigate rare modified nucleoside as promising marker in breast cancer DNA samples. The paper is technically well performed and within the scope of the journal. Nevertheless the Journal scope is mostly based on pharmaceutical chemistry the biological vision plays an important role. In this case also the clinical perspective is important since the analysed DNA samples derive from patients. The technical effort in setting the chemistry parameters to  evaluate of the normal versus tumour tissue are not commensurate to the biological conclusions. It is very difficult to infer the meaning of the subtle differences in rare nucleotide amounts  in  normal versus tumour tissues. More cases have to be analysed.  My main request to the authors  is that they should include  a classical parameter to evaluate the stage of the analysed tumours. In other word compare their result with an widely accepted parameter indicating the staging of the breast tumours. Actually it is unclear whether the presence of rare nucleotide could be prognostic or associated to progression of the tumour. Although apparently this is out of the purpose of the manuscript the authors should add comments

Author Response

Thanks for your comment. To our knowledge, the current staging standard of breast tumours usually uses the TNM staging method which is widely used in the world. In this method, T refers to tumor size, N refers to lymph node metastasis, M refers to distant organ metastasis, and the specific situation of TNM is combined to indicate the staging of breast cancer. In our work, we extracted genomic DNA (gDNA) samples from 20 pairs of breast cancer tissues and the corresponding normal tissues adjacent to the tumor for the quantification of these five modified nucleosides. We found that the contents of 5mdC (p < 0.1), 5hmdC (p < 0.01) and 5fdC (p < 0.1) decreased, while the contents of 5hmdU (p < 0.05) and 5fdU (p < 0.01) increased in tumor tissues compared to normal tissues adjacent to the tumor in general. Since the stage-related information was not involved, we were unable to compare our results with the parameters of the stage. However, it is worth noting that this work found significant changes in the contents of 5hmdC, 5hmdU and 5fdU in breast cancer tissues and adjacent tissues, demonstrating the great potential of 5hmdC, 5hmdU and 5fdU as cancer biomarkers. And simultaneous detection of multiple cancer biomarkers in clinical samples greatly improved the accuracy of cancer diagnosis, indicating that our method had great application potential in clinical multidimensional diagnosis. In the next work, more clinical samples from more breast cancer patients and different types of cancer patients will be analysed to further determine whether these modified nucleosides can become new universal cancer biomarkers for clinical application, which was emphasized in the conclusion part.

Round 2

Reviewer 2 Report

The manuscript has been  improved. It can be accepted